# Anxiolytic-like Effect of Inhaled Cinnamon Essential Oil and Its Main Component Cinnamaldehyde in Animal Models

**DOI:** 10.3390/molecules27227997

**Published:** 2022-11-18

**Authors:** Ly Thi Huong Nguyen, Nhi Phuc Khanh Nguyen, Khoa Nguyen Tran, Heung-Mook Shin, In-Jun Yang

**Affiliations:** Department of Physiology, College of Korean Medicine Dongguk University, Gyeongju 38066, Republic of Korea

**Keywords:** cinnamon essential oil, cinnamaldehyde, anxiety, microarray, electroencephalogram

## Abstract

Aromatherapy is one of the most common safer alternative treatments for psychiatric disorders with fewer side effects than conventional drugs. Here, we investigated the effects of cinnamon essential oil (CIEO) inhalation on mouse behaviors by performing different behavioral tests. CIEO inhalation showed anxiolytic effects in the elevated plus maze test, as inferred from increased time spent in open arms and decreased time spent in closed arms. Moreover, the CIEO treatment enhanced social behavior by increasing the total contact number, time spent in the center, distance traveled in the center, and total distance in the social interaction test. However, CIEO inhalation did not have any effect on performance in the open field test, tail suspension test, forced swimming test, and Y maze tests. The microarray analysis indicated that the CIEO treatment downregulated 17 genes and upregulated 15 genes in the hippocampus. Among them, Dcc, Egr2, and Fos are the most crucial genes that are involved in anxiety-related biological processes and pathways, including the regulation of neuronal death and neuroinflammation. Gas chromatography/mass spectrometry analysis revealed that cinnamaldehyde is the main component of CIEO. Cinnamaldehyde recovered MK-801-induced anxiety-related changes in the electroencephalogram power spectrum in zebrafish. Taken together, our findings suggest that CIEO and its main component cinnamaldehyde have an anxiolytic effect through the regulation of the expression of genes related to neuroinflammatory response and neuronal death.

## 1. Introduction

Anxiety is one of the most common mental disorders and is characterized by excessive and uncontrollable nervousness and fear [1]. Typical subtypes of anxiety include generalized anxiety disorder, obsessive-compulsive disorder, post-traumatic stress disorder, social anxiety disorder, panic disorder (with or without agoraphobia comorbidity), and phobias [2]. Due to the coronavirus disease 2019 (COVID-19) pandemic, the global prevalence of anxiety has increased remarkably, with more than 30% of the general population suffering from this condition [3]. The first-line medications for most anxiety subtypes are selective serotonin (5-HT) reuptake inhibitors (SSRIs), including paroxetine, sertraline, citalopram, escitalopram, fluvoxamine, and fluoxetine [4]. However, the long-term use of these SSRIs is associated with various adverse effects, such as bleeding, digestive problems, hyponatremia, excessive sweating, emotional blunting, and increased risk of suicidality [5,6]. Moreover, several studies have reported delayed onset of action and tachyphylaxis with SSRI treatments [7,8]. These issues raise the need for the investigation and development of alternative medicines for the treatment of anxiety with high efficacy and fewer side effects.

Cinnamon, the bark of *Cinnamomum cassia*, has been widely used in traditional medicine to treat various diseases, such as ischemia, diabetes, peptic ulcer, and cancers [9]. The essential oil derived from cinnamon bark possesses a variety of pharmacological activities, such as antibacterial, antioxidative, and anti-inflammatory effects [10]. Cinnamon essential oil largely contains volatile compounds, such as cinnamaldehyde and its derivatives, which are small molecules that easily penetrate the brain by crossing the blood–brain barrier (BBB) and could be involved in the regulation of the neuroendocrine system and neurotransmitters and thus affect behavioral outcomes [11]. A previous study reported that repeated intraperitoneal injections of cinnamon essential oil exerted anti-depressant and anxiolytic effects in mice [12]. However, the efficacy of cinnamon essential oil inhalation on psychological disorders as well as the underlying mechanism of action has not been elucidated yet.

In this study, we investigated the effects of inhaled cinnamon essential oil (CIEO) on the behaviors of mice and evaluated the gene expression profiles in the hippocampus to investigate the possible underlying mechanisms. The anxiolytic properties of cinnamaldehyde, the main component of cinnamon essential oil, were also assessed using the zebrafish model.

## 2. Results

### 2.1. Effects of CIEO on Anxiety-like Behavior in Mice

An EPM test was conducted to evaluate the anxiolytic effect of CIEO inhalation in mice. Figure 1 shows that the CIEO-treated groups (5% and 10%) exhibited a significant increase in time spent in the open arms and a significant decrease in time spent in the closed arms in the EPM (*p* < 0.05). No differences between groups were observed for the distance traveled in both open and closed arms (Figure 1B).

### 2.2. Effects of CIEO on Locomotor Activity in Mice

The locomotor activity of mice after CIEO inhalation was measured using the OFT. As shown in Figure 2, CIEO did not have any effect on the number of entries, time, and distance in the center, as well as the total distance over a 10-min OFT session.

### 2.3. Effects of CIEO on Social Behavior in Mice

The effect of CIEO on social behavior in mice was assessed by the social interaction test. Inhalation of 5% and 10% CIEO significantly increased the total contact number (*p* < 0.05) between two mice in the social interaction test (Figure 3). In addition, the CIEO5 group showed remarkable increases in the time spent in the center, distance traveled in the center, and total distance (*p* < 0.05), compared to the CON group (Figure 3B).

### 2.4. Effects of CIEO on Depression-like Behavior in Mice

TST and FST were conducted to examine the antidepressant effect of CIEO in mice. Figure 4 indicates that the CIEO inhalation did not have any effect on the immobility time in the TST and FST, compared to the CON mice.

### 2.5. Effects of CIEO on Working Memory in Mice

The effect of inhaled CIEO on working memory was evaluated using the Y maze test. As shown in Figure 5, the CIEO inhalation did not have any effect on the alternation triplet, total arm entries, and total distance in the Y maze test.

### 2.6. Effects of CIEO on Gene Expression in the Hippocampus

To investigate the possible mechanisms for the anxiolytic effect of CIEO, the gene expression profiles in the hippocampus from the CON and CIEO groups were analyzed using microarray. Figure 6A shows the differentially expressed genes (DEGs) between the two groups. The data revealed that there were 17 downregulated genes and 15 upregulated genes (Figure 6A, Appendix A). The GO Biological Process enrichment analysis indicated that the DEGs were associated with several processes, which could be related to neurological diseases, such as the regulation of neuron death (GO:1901214) and positive regulation of myelination (GO:0031643) (Figure 6B). WikiPathway enrichment analysis showed that the DEGs were involved in several pathways related to anxiety, including neuroinflammation WP4919, serotonin, and anxiety-related events WP3944, brain-derived neurotrophic factor (BDNF) signaling pathway WP2380, and serotonin and anxiety WP3947 (Figure 6C). Dcc, Egr2, and Fos are the crucial DEGs that are involved in these biological processes and pathways (Appendix A).

### 2.7. Chemical Composition of CIEO

The chemical composition of CIEO was analyzed using gas chromatography/mass spectrometry (GC/MS). The chromatogram is shown in Figure 7A. The identified components of CIEO and their relative contents are indicated in Figure 7B. There were 27 volatile compounds detected in CIEO and (*E*)-cinnamaldehyde was the most abundant compound in CIEO (82.74%).

### 2.8. Effects of Cinnamaldehyde on EEG in an MK-801-Induced Anxiety-like Model in Zebrafish

Next, the anxiolytic effects of cinnamaldehyde, the main component of CIEO, in the MK-801-induced anxiety-like model in zebrafish were investigated. Compared to the CON group, the MK-801 group showed significant increases in the relative power spectral densities of slow oscillation, pure delta, and delta (*p* < 0.05), and significant decreases in the relative power spectral densities of beta2, beta, and slow gamma (*p* < 0.05) (Figure 8A). These changes were remarkably reversed by cinnamaldehyde treatment (5 mg/L) (*p* < 0.05). As shown in Figure 8B, the cinnamaldehyde exposure significantly reduced MK-801-induced increases in delta/beta ratio (DBR) and theta/beta ratio (TBR) in zebrafish (*p* < 0.05). The effects of cinnamaldehyde were comparable with the positive control, haloperidol (9 μM).

## 3. Discussion

Aromatherapy has been one of the most common complementary and alternative treatments for psychological diseases, such as sleep disorders, anxiety, and depression [13]. Essential oils have high efficacy with fewer side effects as compared to conventional drugs [14]. However, the underlying mechanisms that mediate the beneficial effects of essential oils in the treatment of psychiatric disorders remain to be elucidated. This study indicated the anxiolytic effect of cinnamon essential oil (CIEO) and its main component, cinnamaldehyde, in animal models and suggested the possible mechanisms of action.

In the present study, the mice were allowed to inhale CIEO, and the effects were evaluated using behavioral tests. Inhalation is the most common application method for clinical trials using essential oils because it is simple and convenient without the requirement of any special equipment. The inhalation method was thought to have a lower risk of side effects and act more rapidly than other methods of drug administration [15]. In this study, we demonstrated that CIEO inhalation for 1 h exerted anxiolytic-like effects in the EPM test, as inferred from increased time in the open arms and reduced time in the closed arms. In a previous report, repeated intraperitoneal injection of CIEO for 14 days also exhibited beneficial effects by alleviating anxiety in mice [12]. These results suggest that the inhalation route of administration of CIEO could have a more rapid anxiolytic effect as compared to systemic administration. This might due to the fact that drugs administered by the intraperitoneal route are absorbed primarily through the portal vein and must pass through the liver before reaching systemic circulation and the brain [16]. In contrast, inhaled essential oils not only act directly on the olfactory system but also easily cross the blood–brain barrier to affect the production of neurotransmitters and the activity of the neuroendocrine system, to modulate behavioral outcomes [17,18]. Moreover, volatile compounds in essential oils might be absorbed rapidly in the respiratory system due to the large surface area of the respiratory endothelium, and easily penetrate the bloodstream to reach the brain [19]. Despite the anxiolytic effects seen in the EPM test, inhaled CIEO did not have any impact on locomotor activity in the OFT which is a common behavioral test to measure locomotor and anxiety-like behaviors. However, OFT is considered less specific than the EPM test in the assessment of anxiety [20]. Our results are in line with previous studies that have also demonstrated the anxiolytic effects of several essential oils and volatile compounds in the EPM test, but not in the OFT [21,22,23]. The social interaction test revealed that when CIEO-treated mice were exposed to the same arenas as in the OFT but paired with an unfamiliar mouse (same treatment), they showed significantly higher social interaction and locomotor activity, suggesting that CIEO might have beneficial effects on social anxiety disorder, a subtype of anxiety. Our results showed that the CIEO treatment had no effect on depression-like behavior in the FST and TST, as well as on working memory in the Y maze test. This could be due to the longer period required for essential oil treatment to exert antidepressant and memory-improving effects [12,24,25]. An earlier study showed that inhaled lavender essential oil (LAEO) significantly reduces anxiety-like behavior in mice by regulating serotonergic transmission [26]. Hence, LAEO was used as a positive control in this study. Interestingly, in the social interaction test, although both CIEO and LAEO could notably increase the total contact number of mice, only CIEO elevated locomotor activity in the center of the arena, indicating its superior effect on social anxiety disorder compared to LAEO.

Microarray analysis revealed that CIEO inhalation downregulated 17 genes and upregulated 15 genes in the hippocampus, in comparison with the CON mice. Among them, Dcc, Egr2, and Fos play an important role in anxiety-related biological processes and pathways, and our study indicates that the anxiolytic effects of CIEO could be mediated through the regulation of these genes. Dcc (or DCC, deleted in colon carcinoma) gene encodes a transmembrane protein, which acts as a receptor of neutrin-1, a key protein in axonal guidance and neuronal cell death induction [27]. A previous study indicated that the DCC gene expression showed a strong positive correlation with anxiety-like behavior in marmosets [28]. The Egr2 (or EGR2, early growth response 2) gene encodes a transcription factor regulating the expression of target genes, which play crucial roles in the myelination process [29]. Egr2 is highly expressed in neurons and is involved in the regulation of neuroinflammation [30]. The expression of EGR2 was associated with stress response and psychiatric disorders [31,32]. The Fos gene (also known as c-fos) is the gene encoding Fos protein, a subunit of transcription factor AP-1 (activator protein 1). Fos is expressed in neurons upon depolarization and is known as a neural activity marker [33]. The expression of c-fos at the mRNA level in the hippocampus, amygdala, and cortex was significantly elevated after exposure to an acute anxiogenic stimulus (EPM test) [34]. In a prior study, the deficiency of c-fos resulted in reduced anxiety-like behavior in mice [35]. Moreover, c-Fos plays a role in regulating neuroinflammation and apoptosis [36,37]. In the current study, we found that the expressions of Dcc, Egr2, and Fos genes were down-regulated in the hippocampus of CIEO-treated mice, suggesting that the anxiolytic effect of CIEO could be through regulating the expression of these genes.

To provide more convincing evidence, a model of zebrafish was used in the present study, in addition to the mouse model. Zebrafish are rapidly becoming a promising in vivo model for neuroscience research, including anxiety and stress studies, due to their neurotransmitter systems, receptors, and hormones [38,39,40]. Moreover, the zebrafish model exhibited behavioral changes in response to stress and showed high predictive validity for psychiatric drug development [41,42]. In this study, we examined the anxiolytic effects of cinnamaldehyde on the EEG signals of MK-801-induced anxiety in zebrafish. GC/MS analysis revealed that cinnamaldehyde is the main bioactive compound in CIEO (82.74%). Cinnamaldehyde is a common compound with various pharmacological properties, such as anti-inflammatory, antibacterial, and anti-cancer effects [43,44,45]. MK-801 is an N-methyl-D-aspartate (NMDA)-type glutamate receptor antagonist and exposure to MK-801 resulted in anxiety-like behavior and reduced social interaction behavior in adult zebrafish [46]. Five typical frequency bands of EEG include delta (0.5–4 Hz), theta (4–8 Hz), alpha (8–12 Hz), beta (12–30 Hz), and gamma (>30 Hz) [47]. It has been demonstrated that psychiatric disorders are associated with alterations in EEG signal, particularly increases in the densities of delta and theta waves and decreases in the densities of alpha, beta, and gamma waves [48]. In addition, the theta/beta ratio and delta/beta ratio had positive correlations with anxiety-like behaviors [49,50]. In this study, the MK-801-induced anxiety model in zebrafish exhibited higher relative power densities of the delta band and lower relative power densities of the beta and slow gamma bands. In contrast, cinnamaldehyde treatment recovered all these changes in the EEG power spectrum as well as reducing TBR and DBR induced by MK-801. These results suggest that cinnamaldehyde displays anxiolytic effects in the zebrafish model. Previous studies have implied that cinnamaldehyde could have therapeutic effects on psychiatric disorders. This is due to its neuroprotective and anti-neuroinflammatory effects in both in vitro and in vivo models [51,52,53]. These data are in line with the results of our study that CIEO, an essential oil containing cinnamaldehyde, reduces the expression of genes related to neuronal cell death and neuroinflammatory pathways. Taken together, the present study has demonstrated the anxiolytic effect of CIEO with cinnamaldehyde as its main component through the regulation of neuronal death and neuroinflammatory response.

## 4. Materials and Methods

### 4.1. Preparation of Cinnamon Essential Oil

Cinnamon bark (50 g) was immersed in 500 mL of distilled water in a 1000 mL flask. Then, 15 mL of n-hexane was used as the extraction solvent and was placed in a 100 mL flask. The extraction of cinnamon essential oil was performed using a simultaneous distillation-solvent extraction (SDE) apparatus for 4 h. The distilled oil was captured in 15 mL of n-hexane, and dried with anhydrous sodium sulfate (Na_2_SO_4_). The essential oil and hexane were separated using a rotary evaporator. Subsequently, the obtained essential oil was weighed (yield 0.75% *w*/*w*) and stored at –20 °C.

### 4.2. Gas Chromatography-Mass Spectrometry (GC-MS) Analysis

The components of cinnamon essential oil (CIEO) were analyzed using a GC-MS system (Agilent, CA, USA) with a DB-5ms column (30 m × 250 μm, 0.25 μm thickness). The oven temperature was set as follows: 40 °C for 5 min, then increased at 2 °C/min to 280 °C (10 min isothermal). The helium gas flow was 1.0 mL/min with a split ratio of 1:20. The temperatures of the inlet and detector were maintained at 250 °C and 280 °C, respectively. The scan mode was from 50 to 500 amu. The sample (1 μL, 518.25 mg/mL) was injected into the column for analysis. The components of CIEO were determined by comparing them with a mass spectra library (National Institute of Standards and Technology [NIST] Mass Spectral Database).

### 4.3. Animal and Treatment with Cinnamon Essential Oil

Five-week-old male C57BL/6 mice were purchased from Koatech (Gyeonggi, South Korea) and allowed to acclimatize for two weeks before the initiation of the experiments. All animal experimental processes were approved by the Institutional Animal Care and Use Committee (IACUC) of Dongguk University, South Korea (Approval no. IACUC-2021-15). The mice were assigned to four groups (*n* = 8 per group): the control group (CON), 5% CIEO-treated group (CIEO5), 10% CIEO-treated group (CIEO10), and 5% lavender essential oil-treated group (LAEO). A piece of filter paper (ϕ110 mm) was immersed with 1 mL of essential oils (dissolved in 3% Tween 80 in saline) or 1 mL of the vehicle. The inhalation of essential oils was performed by placing the filter paper immersed with the essential oils or the vehicle in a plastic cage (24 cm × 39 cm × 19 cm) containing the mouse for 1 h before behavioral testing.

### 4.4. Elevated plus Maze (EPM) Test

To examine the anxiolytic effects of CIEO mice, an EPM test was performed as previously described [54]. All mice were habituated to the test room for 2 h before the test. The mice were put in the central area of a plus maze (arm length: 30 cm, arm width: 5 cm) and were free to move in the maze for 5 min. The entry number, time, and distance in the open arms were measured using the SMART v3.0 tracking software (Panlab, Barcelona, Spain).

### 4.5. Open Field Test (OFT)

An OFT was performed to investigate the effects of CIEO on locomotor activity in mice as previously described [55]. All mice were habituated to the test room for 2 h before the test. The mice were put in the center of a box (30 cm × 30 cm × 30 cm) and were free to move in the box for 10 min. The entry number, time (s), and distance (cm) in the central zone, as well as the total distance (cm), were measured using the SMART v3.0 tracking software (Panlab, Barcelona, Spain).

### 4.6. Social Interaction Test

The social interaction test was conducted as previously described with some modifications [56]. All mice were habituated to the test room for 2 h before the test. Mice were assigned in pairs with the mice from unfamiliar home cages and the same treatment. The mice were put in the opposite corners of a box (30 cm × 30 cm × 30 cm). The total contact numbers, time spent in the center (s), distance traveled in the center (cm), and total distance (cm) were recorded in a 5-min session with the SMART v3.0 tracking software (Panlab, Barcelona, Spain).

### 4.7. Tail Suspension Test (TST)

To examine the antidepressant-like effects of CIEO in mice, TST was performed as previously described [57]. All mice were habituated to the test room for 2 h before the test. Briefly, the mouse was hung to a bar (50 cm above the ground) using tape (15 cm long) attached to the tail. The immobility time (s) of the mice was measured in a 6-min session with the SMART v3.0 tracking software (Panlab, Barcelona, Spain).

### 4.8. Forced Swimming Test (FST)

The FST was conducted to investigate the effects of CIEO on depression in mice as previously described [58]. All mice were habituated to the test room for 2 h before the test. Briefly, the mouse was put in a cylinder (height: 30 cm, diameter: 20 cm) filled with 10 cm of water (25 °C). The total immobility time (s) of the mice was assessed in a 6-min session with the SMART v3.0 tracking software (Panlab, Barcelona, Spain).

### 4.9. Y Maze Test

The Y maze test was conducted to evaluate the effects of CIEO on working memory and exploratory activity as previously described [59]. All mice were habituated to the test room for 2 h before the Y maze test. The mice were put in the center of the Y maze (arm length: 40 cm, wall height: 10 cm). The total arm entries, alteration triplet (%), and total distance (cm) were recorded for 10 min with the SMART v3.0 video tracking system (Panlab, Barcelona, Spain).

### 4.10. Sample Collection and Microarray Analysis

On the day following the final behavioral test, the mice were sacrificed by using isoflurane inhalation. The hippocampus was dissected and stored in RNAlater solution (Thermo Fisher Scientific Baltics, Vilnius, Lithuania) for further analysis. RNA from the hippocampus was extracted using a TRIzol reagent (Sigma Aldrich, St. Louis, MO, USA). Microarray analysis for samples from the CON and CIEO5 groups was conducted using SurePrint G3 Mouse Gene Expression 8 × 60 K (Agilent, Inc., Santa Clara, CA, USA), according to the manufacturer’s instructions. Labeled cRNA was prepared from total RNA using Agilent’s Quick Amp Labeling Kit, then hybridized to the Agilent Expression Microarray according to the protocols provided by the manufacturer. Arrays were scanned and analyzed using the Agilent Technologies G4900DA SG12494263 and Agilent Feature Extraction (v11.0.1.1). Genes with a fold change >1.5 or <−1.5 and *p*-value < 0.05 (independent Student’s *t*-test) were determined as differentially expressed genes (DEGs). The Enrichr tool was used for the enrichment analysis of DEGs with the Gene Ontology (GO) Biological Process and WikiPathway database [60].

### 4.11. Electroencephalogram (EEG) Signal in Zebrafish

Adult zebrafish (wild-type, AB strain) were maintained at Zefit Co., Ltd., (Daegu, South Korea) under a 14 h:10 h light: dark cycle. The water for animal maintenance was purified using the reverse osmosis method and kept at a pH of 6.5–7.5 and 27 ± 1 °C. The adult zebrafish were placed in a 300 mL water bath containing 16 mg/L (*v*/*v*) of eugenol to induce stage 3 anesthesia. MK-801 (M107, Sigma-Aldrich, St. Louis, MO, USA) was used to induce an anxiety-like model in the zebrafish as previously described [17]. After 30 min of pre-treatment with MK-801, the zebrafish were treated with (*E*)-cinnamaldehyde (5 mg/L) (CFN90321, Chemfaces, Wuhan, China) or haloperidol (a positive control drug, 9 μM) for another 30 min and their EEG was measured for 20 min using non-invasive electrodes. An MP36 device (Biopac Systems Inc., Goleta, CA, USA) was used to measure and process the EEG signals. The fast Fourier transform method was used to analyze the frequency of the signals. Relative power spectral densities (slow oscillation, pure delta, delta, theta, alpha, beta1, beta2, beta, and slow gamma) were recorded. The delta/beta and theta/beta ratios were considered anxiety-related markers.

### 4.12. Statistical Analysis

GraphPad Prism 9.0 (GraphPad Software, San Diego, CA, USA) was used for statistical analysis. The results are presented as means ± standard deviation (SD) of at least three independent experiments. Differences between groups were analyzed using one-way ANOVA or two-tailed unpaired Student’s *t*-test and *p*-values < 0.05 were considered statistically significant.

## 5. Conclusions

The present study demonstrated the anxiolytic effect of cinnamon essential oil and its major component cinnamaldehyde in mouse and zebrafish models. The microarray analysis suggests that the underlying mechanisms of the anxiolytic effects could be through the regulation of the expression of genes related to neuronal cell death and neuroinflammatory pathways. Future research should consider the antidepressant potential of cinnamon essential oil in a longer duration treatment as well as its long-lasting anxiolytic-like effects. Moreover, the exact mechanisms underlying the anti-anxiety effects of cinnamon essential oil should be validated.

## Figures and Tables

**Figure 1 molecules-27-07997-f001:**
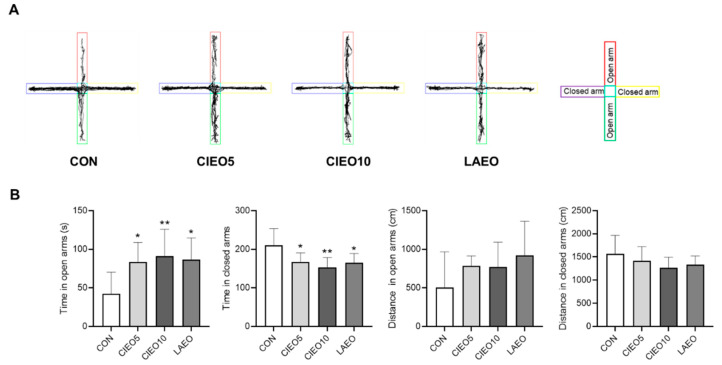
Effect of CIEO on anxiety-like behavior in mice. (**A**) Traveling paths in the EPM test. (**B**) The time and distance in the open arms and closed arms in the EPM were measured in a 5-min session. Data represent means ± SDs (*n* = 8 per group). * *p*  <  0.05, ** *p*  <  0.01 vs. the CON group. CON: control; CIEO: cinnamon essential oil; LAEO: lavender essential oil.

**Figure 2 molecules-27-07997-f002:**
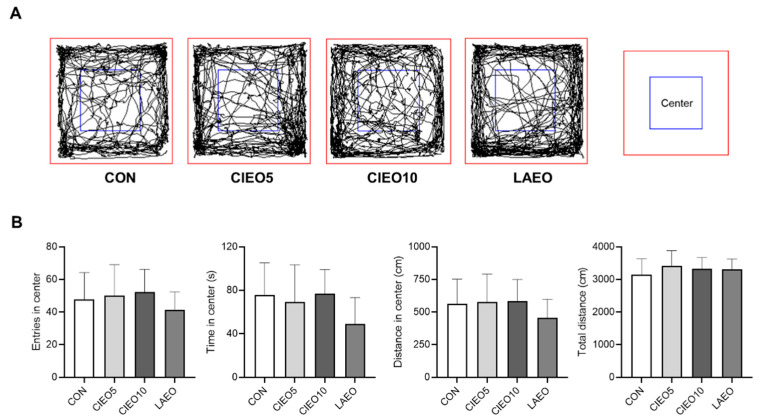
Effect of CIEO on locomotor activity in mice. (**A**) Traveling paths in the OFT. (**B**) The entry number, time, distance in the center, and total distance were recorded in a 10-min session. Data represent means ± SDs (*n* = 8 per group). CON: control; CIEO: cinnamon essential oil; LAEO: lavender essential oil; OFT: open field test.

**Figure 3 molecules-27-07997-f003:**
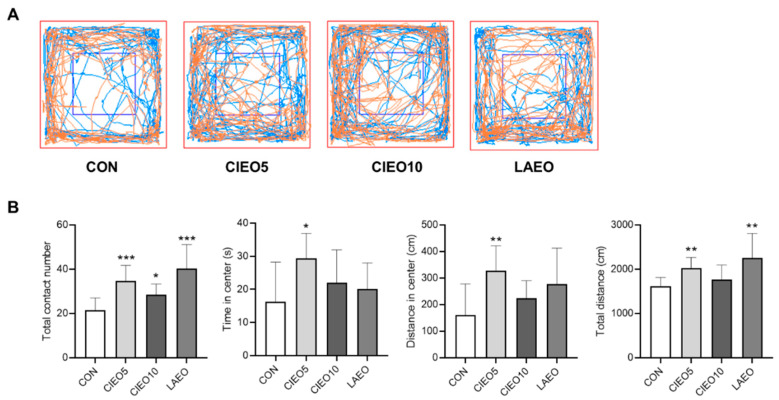
Effect of CIEO on social behavior in mice. (**A**) Traveling paths in the social interaction test. (**B**) The total contact number, time spent in the center, distance traveled in the center, and total distance were recorded in a 5-min session. Data represent means ± SDs (*n* = 8 per group). * *p*  <  0.05, ** *p*  <  0.01, *** *p*  <  0.001 vs. the CON group. CON: control; CIEO: cinnamon essential oil; LAEO: lavender essential oil.

**Figure 4 molecules-27-07997-f004:**
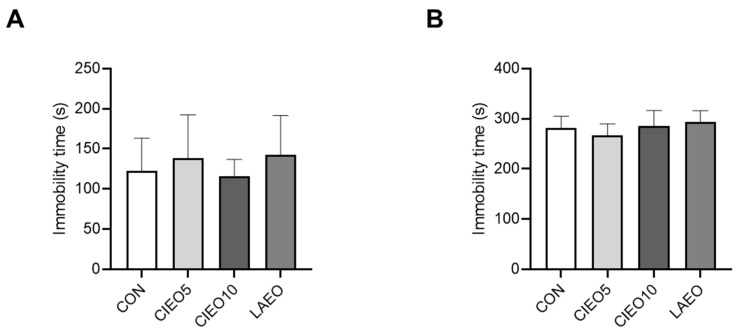
Effect of CIEO on depression-like behavior in mice. (**A**) The immobility time in a 6-min session of TST was recorded. (**B**) The immobility time in a 6-min session of FST was measured. Data represent means ± SDs (*n* = 8 per group). CON: control; CIEO: cinnamon essential oil; LAEO: lavender essential oil; TST: tail suspension test; FST: forced swimming test.

**Figure 5 molecules-27-07997-f005:**
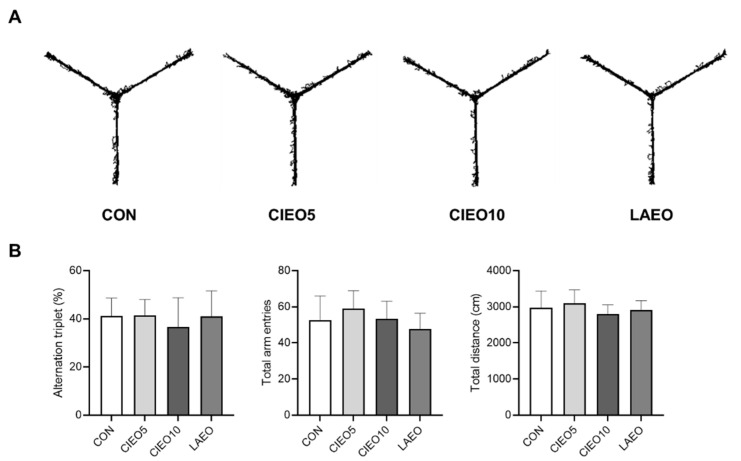
Effect of CIEO on working memory in mice. (**A**) Representative traveling paths in the Y maze test. (**B**) The alternation triplet, total arm entries, and total distance were recorded in a 10-min session. Data represent means ± SDs (*n* = 8 per group). CON: control; CIEO: cinnamon essential oil; LAEO: lavender essential oil.

**Figure 6 molecules-27-07997-f006:**
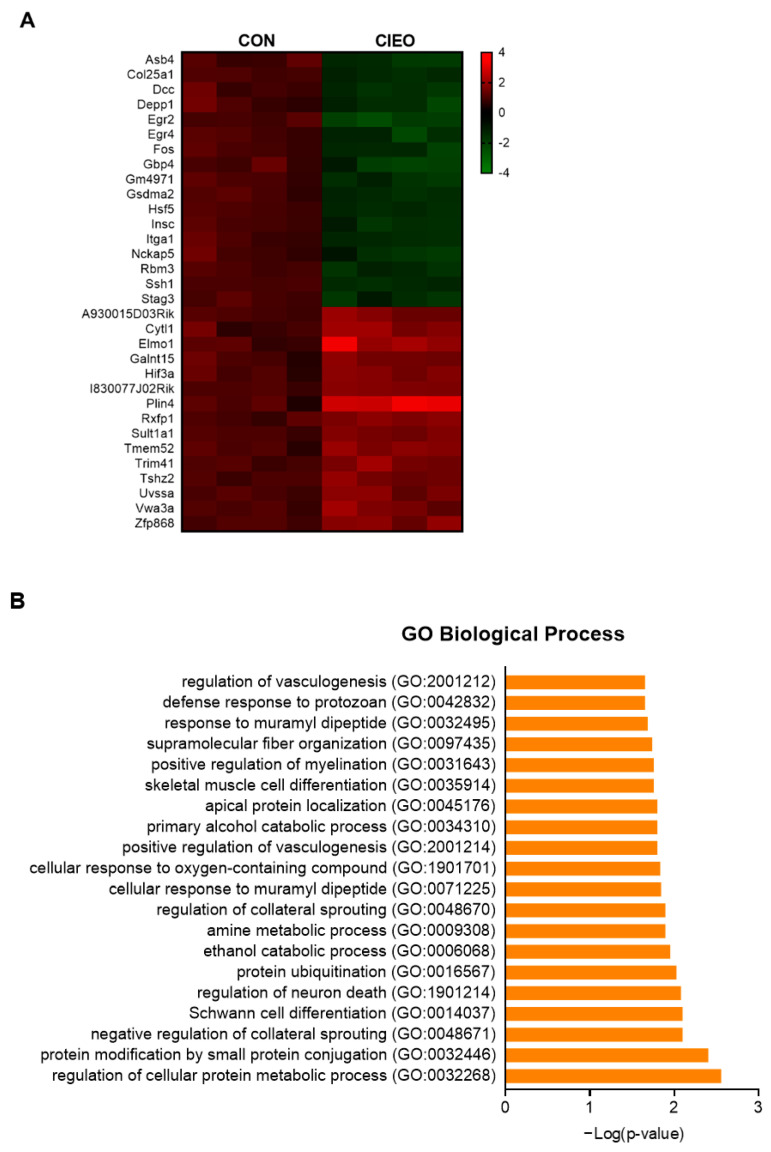
Effects of CIEO on gene expression in the hippocampus. (**A**) Heatmap representing the relative expression of DEGs in the hippocampus between the CON group and the CIEO group. (**B**) GO Biological Process enrichment of DEGs. (**C**) WikiPathway enrichment analysis of DEGs. DEG: differentially expressed genes; GO: Gene Ontology.

**Figure 7 molecules-27-07997-f007:**
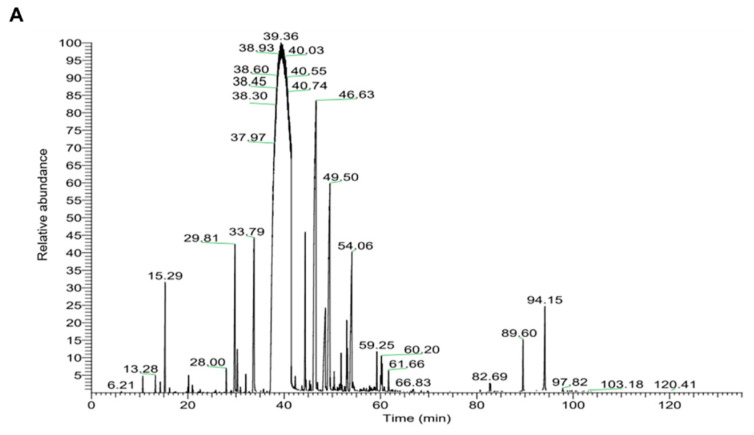
Chemical composition of CIEO. (**A**) GC/MS chromatogram of CIEO. (**B**) Retention time and relative content of detected compounds in CIEO. GC/MS: Gas chromatography/mass spectrometry.

**Figure 8 molecules-27-07997-f008:**
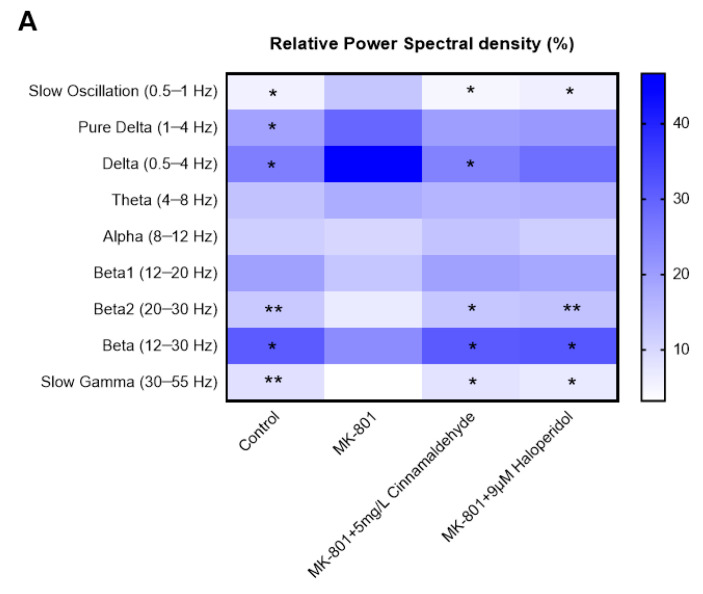
Effects of cinnamaldehyde on EEG in an MK-801-induced anxiety-like model in zebrafish. (**A**) Heat map representing EEG relative power spectrum. (**B**) Theta/beta and delta/beta ratios were calculated. EEG: electroencephalogram. Data represent means ± SDs (*n* = 6 per group). * *p* < 0.05, ** *p* < 0.01 vs. the MK-801 group.

## Data Availability

The data presented in this study are available in this article.

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
