# Peer review of "Anxiolytic-like Effect of Inhaled Cinnamon Essential Oil and Its Main Component Cinnamaldehyde in Animal Models"

_molecules, 2022, doi:10.3390/molecules27227997_

Round 1
Reviewer 1 Report
Please revise with reference to the attached document.

Author Response
[Comment 1] P5L112: You only compared the control and cinnamon essential oil, why didn't you examine the changes in gene expression levels under lavender and different concentrations of cinnamon essential oils as well? Also, please describe which concentration of cinnamon essential oil you used for this analysis?
Response: Thank you for pointing this out. In the EPM test, all cinnamon and lavender essential oil treatments showed comparable anxiolytic effects. However, in the social interaction test, the 5% cinnamon essential oil (CIEO5) treatment exerted more beneficial effects on social anxiety disorder (as inferred from time in the center, distance in the center, and total distance). Lavender oil treatment in this study plays a role as a positive control, hence, we did not center on it. This is the reason why the group treated with 5% cinnamon essential oil was used for microarray analysis. Information about that concentration was already mentioned in section 4.10.
[Comment 2] P10 L162: You compared the effects of cinnamon essential oil and lavender essential oil in the behavioral analysis, is there any need to mention them in the discussion?
Response: Thank you for your suggestion. Further discussion about this issue has been implemented in the manuscript (P11 L201-206).
[Comment 3] P10 L181: Please describe the mechanism by which the inhalation was faster effect than the intraperitoneal administration.
Response: Thank you for your recommendation. We described the possible mechanism in the Discussion part (P10 L180-188).
[Comment 4] P11 L222: Nowhere in this paper is there a reason to examine using the different animals. Also, the sample provided to the mice is an essential oil and to the zebrafish is a single compound. The reason for this is not described. Furthermore, essential oils are a complex of components, and some of them have strong effects even if their content is small. Please explain why you have chosen to focus only on cinnamon aldehyde as the basis for the functionality of cinnamon essential oil.
Response: Thank you for your questions. Further explanation for using zebrafish in addition to mice was added in the Discussion part (P11 L229-235). In this study, we focused only on cinnamaldehyde due to several reasons. First, it is the main active compound of cinnamon essential oil, accounting for over 80% (data from GC/MS analysis). Moreover, cinnamaldehyde has been reported to exhibit anxiolytic effects on mice behaviors (in both EPM and OFT), hence, in the current study, we examined the anxiolytic effect of cinnamaldehyde, not other minor compounds, in a different in vivo model of anxiety (MK-801-treated zebrafish).
[Comment 5] P12L247: The description regarding the preparation of the materials is vague. Is the extraction method used in this study a combination of steam distillation and solvent extraction? In addition, it should be clear whether your materials are essential oils or extracts, please describe the technical words and preparation method in an organized manner.
Response: Thank you for your suggestion. In this study, the cinnamon essential oil was prepared using steam distillation, and n-hexane was an organic solvent to capture the essential oil. We have modified the description to make it explicit.
[Comment 6] P12L268: Is the essential oil diluent the same composition of vehicle as the control? Also, please describe how long after the addition of the test sample, the mice were placed in their cages.
Response: Thank you for your suggestion. Yes, the same composition of vehicle (3% Tween 80 in saline) was used for the CON sample and essential oil dilution. We have modified the description to make it explicit.
[Comment 7] P13L332: Is the cinnamaldehyde used (E) or (Z) type? Also, please indicate the manufacturer and model number of the compound. Similarly, please provide appropriate information on MK-801. Also, what is the rationale for setting the test quantity of cinnamaldehyde at 5 mg/L?
Response: Thank you for your comment. The cinnamaldehyde used in this study is (E) type. We added the information of cinnamaldehyde and MK-801 in section 4.11. The dose of 5 mg/L of cinnamaldehyde was chosen following a previous study that other essential oil compounds (citral and geraniol) at 1, 5, and 10 mg/L were used to investigate anxiolytic effects in the zebrafish model (J Ethnopharmacol 2020, 260:113036).
[Comment 8] P14L341: Please indicate exactly what analysis method was used for which data.
Response: Thank you for your comment. We added the analysis method for this data.
[Comment 9] Others, such as (E), p-, etc. should be italicized. Please check again and describe them correctly.
Response: Thank you very much for your suggestion. We have checked and corrected them.
Reviewer 2 Report
1. Provide the % yield of the cinnamon essential oil obtained after distillation
2. Did you optimize the experiments such as EPM, OFT, TST, social interaction test, FST, and Y maze test? Kindly describe. Was a preliminary testing/ trial performed to train the mice and the observers?
3. Indicate the number of trials performed per test.
4. The authors may include the treatment of the animals for 1 - 2 wks ( inhalation per day) as future directions of the study to observe an effect in the animal tests as what is stated in the discussion, p. 11 line 196
Author Response
[Comment 1] Provide the % yield of the cinnamon essential oil obtained after distillation
Response: Thank you for your suggestion. We have added the required information.
[Comment 2] Did you optimize the experiments such as EPM, OFT, TST, social interaction test, FST, and Y maze test? Kindly describe. Was a preliminary testing/ trial performed to train the mice and the observers?
Response: Thank you for your comment. However, we conducted the behavioral tests following the protocols described in previous studies and we did not optimize and perform preliminary testing to train the mice and the observer. In these protocols, the mice were not trained before the tests because memories after preliminary testing might affect the behavioral outcomes. All mice were habituated to the test room for 2 h before the test. All the tests were automatically recorded and analyzed using the SMART v3.0 video tracking system and the observer had previous experience in conducting behavioral tests in mice with several publications (Evid Based Complement Alternat Med 2021, 2021:6687513; Molecules 2022, 27(4), 1412; Evid Based Complement Alternat Med 2022, 2022:1307173).
[Comment 3] Indicate the number of trials performed per test.
Response: Thank you for your suggestion. We have added the required information.
[Comment 4] The authors may include the treatment of the animals for 1 - 2 wks (inhalation per day) as future directions of the study to observe an effect in the animal tests as what is stated in the discussion, p. 11 line 196.
Response: Thank you for your great suggestion. We have added this information in the Conclusion section.
Reviewer 3 Report
The manuscript is definitely interesting. Can be accepted in its current form.
Author Response
[Comment] The manuscript is definitely interesting. Can be accepted in its current form.
Response: Thank you very much for your comment.
Round 2
Reviewer 1 Report
I confirmed that you have revised accordingly according to my comments. Thank you very much. I look forward to your further research.